# Numerical Study of the Reinforcing Pads Geometry of Pressure Vessels

**DOI:** 10.3390/ma18102318

**Published:** 2025-05-16

**Authors:** Grzegorz Świt, Michał Szczecina

**Affiliations:** Department of Civil Engineering and Architecture, Kielce University of Technology, 25-314 Kielce, Poland; gswit@tu.kielce.pl

**Keywords:** steel, pressure vessels, finite element method, reinforcing pads

## Abstract

The structural design of pressure vessels is a rather complicated engineering task and demands on using finite element method (FEM) software to recreate many issues accompanying the design process. One of them is a choice of the shape of reinforcing pads, connecting a min shell of a vessel with nozzles. The mentioned issue is very rarely taken up by researchers. Some of them considered different reinforcement pads (circular and elliptical) using the finite element method (FEM), but they presented results of a nozzle-shell connection without describing results for the rest of the vessel. The other authors performed a thorough FEM analysis of a vessel, but they considered only circular reinforcing pads. The authors of this paper analyzed a pressure vessel without and with reinforcing plates with an elliptical shape. They performed FEM calculations of the vessels using a non-linear material model and a coupled thermal-stress analysis in Abaqus software. The use of the elliptic plate resulted in a considerable decrease in the thickness of the shell and turned out to be an interesting alternative to circular pads. In the presented example, the percentage decrease in thickness was equal to 36%, and the total mass savings was 30%.

## 1. Introduction

The structural design of pressure vessels for gases is a very specific, practical, and scientific issue. Nowadays, not only ultimate and serviceability limit states, but also carbon footprint and energy savings should be taken into account while designing such structures. It is crucial to optimize the shell thickness of such structures. The authors of this paper present new results of the finite element method (FEM) analysis of pressure vessels. The novelty of the presented work is the possibility to decrease shell thickness using elliptical reinforcing pads in the nozzle-shell connections. So far, the circular reinforcing pads have been the most commonly used in practice. Contemporary scientific works on the pressure vessel can be divided into a few groups of issues, among others: review papers, FEM analysis, optimization, analysis of the main shell, analysis of the nozzle-shell connection.

The structural design of the pressure vessels is a well-described issue, both in codes and handbooks. It is also a quite popular subject of scientific research. In the late 1960s, Detman [1] proposed to develop analysis methods to formulate structural design for elevated temperatures, creep behavior with time-dependent stress history combined with relaxation behavior, fatigue, and plastic behavior of the vessels. Patil and Jadhav [2] reviewed 18 important papers on the failure and optimization of pressure vessels. Reinforcing pads were mentioned in one paper only. Toudehdehghan and Hong [3] provided an overview of the pressure vessels, describing their classification, applications, materials, failures, loadings, codes, and standards.

Many papers concerning FEM as an effective tool of advanced analysis of the vessels appeared at the turn of the 20th and 21st centuries. Singh [4] presented a FEM calculation of stress in welded reinforcing pads and applied two kinds of finite elements: an axisymmetric quadratic solid element and a 3-node thick shell element, but the paper presents no details of the FEM model. Sang et al. [5] performed laboratory tests and FEM calculations (using ADINA software) of inelastic stress for a vessel-nozzle intersection, but without a reinforcing pad and presenting a nozzle-shell connection only (no results for the rest of the main shell). Miranda et al. [6] employed ANSYS software (ver. 9.0, Ansys Inc. Canonsburg, PA, USA) to calculate a stress distribution in the vessel-nozzle intersection. Moreover, they considered non-reinforced and reinforced intersections and compared FEM results with an experiment. The FEM analysis took into consideration a bonded and a frictional interaction between the vessel and the nozzle, but the authors analyzed the circular openings and pads only. Fang et al. [7] proved an usefulness of the reinforcing pads by laboratory tests and FEM calculations, but once again the pads were circular. Jamadar et al. [8] showed a use of a non-linear FEM (CATIA and ANSYS) and calculated stress in a shell and at the nozzle-vessel intersection, but once again only circular pads were considered. Maghrabi et al. [9] demonstrated the elastic behavior of a vessel with a lateral nozzle (with different lateral angles) using ANSYS software, but only in an elastic range. Heng et al. [10] presented how to design a vessel with Midas NFX software, taking into account pressure and temperature, but without a broader analysis of the shell-nozzle connection. Balac et al. [11] investigated crack growth on a vessel surface and in nozzles using XFEM, but the nozzles were not equipped with reinforcing pads. Jain et al. [12] used ANSYS software to analyze a pressure vessel both with and without reinforcing pads. A large reduction of maximal stress and deformation in the intersection of the vessel and nozzles was observed, but only in the case of circular pads. Barjod and Shah [13] also performed FEM calculations using ANSYS but focused on an influence of an offset and an inclination angle of a nozzle on obtained results. The minimal von Mises stress was observed in the case of the 32-inch offset and the 0° inclination angle. Once again, the model contained no reinforcing pads. Abdalla [14] generated plastic collapse moment boundaries using a few criteria: the plastic work curvature, the plastic work, and the twice elastic slope method. The author took into consideration vessels with and without reinforcing pads and also compared numerical results with laboratory tests; however, only circular pads were considered.

Contemporary papers still present a use of various software in the FEM analysis of the vessels, for instance:ANSYS–Salins et al. [15], Pany [16], Rapeta et al. [17], Biradar [18],Abaqus–Stikvoort [19],Catia–Chowdary and Paul [20],Inventor–Haziziand Ghaleeh [21].

Not only FEM analyses but also the optimization of vessels is a current scientific issue. Reuss [22] showed how to perform a value analysis of a vessel and presented a few creative ideas to improve design. Carbonari et al. [23] discussed the shape optimization of a vessel using a multi-objective function to minimize the von Mises stress from a nozzle to the head of the vessel. Nasseri et al. [24] used geometric programming to optimize the costs (material, forming, and welding) of a cylindrical vessel with hemispherical heads. Gupta and Desai [25] pondered different nozzle offsets and inclination angles to find an optimal location of a nozzle and used ANSYS software to perform the analysis. Hassan et al. [26] optimized a pressure vessel using a metaheuristic approach (ant colony optimization algorithm) and checked the obtained results using ANSYS software. Ke et al. [27] presented the so-called fruit fly algorithm and showed how to use it to optimize the total cost of a vessel. Widiharso et al. [28] carried out an optimization of a pressure vessel using ANSYS software with the following optimization variables: wall thickness, local von Mises stress, and mass. Woldemichael and Woldeyohannes [29] proposed how to use an open-source pyOpt framework for non-linear optimization problems. They demonstrated the framework performing the cost optimization of a pressure vessel. There are also other known optimization techniques, for instance, the Real Coded Genetic Algorithm presented by Nishidhar Babu et al. [30] and the genetic algorithm demonstrated by Imran et al. [31]. Finally, there are also works considering the optimization of a flat-end head of pressure vessels, presented by Romanowicz and Szybiński [32,33]. All the abovementioned works [22,23,24,25,26,27,28,29,30,31,32,33] did not take reinforcing pads into consideration as a very important variable in the optimization process. The optimization process was devoted to the main shell of a vessel.

The elliptical reinforcing pads are barely discussed in scientific papers. Faisal et al. [34] compared a circular and an elliptical reinforcing pad and stated that the reduction in stress in a nozzle-shell intersection is slightly higher in the case of the circular pad, but the elliptical pad is lighter and therefore less expensive. However, the authors considered only a stress distribution in the nozzle-shell intersection without analyzing the whole vessel. Ismail and Ghazali [35] presented a literature survey on the development of a reinforcing pad design, taking into account laboratory tests and numerical analyses, but the presented papers discussed circular pads only.

Recently, a few papers on pressure vessels have appeared, but they are devoted mainly to fatigue analysis of the reactor pressure vessels [36,37] or take into consideration oblique nozzles with reinforcing pads [38], while the authors of this paper analyzed nozzles perpendicular to the shell only and the pressure vessels under static loadings.

As we can see, the idea of the use of the elliptical pads needs a broader discussion and analysis. Moreover, most authors perform linear FEM analysis using an elastic material model. They also rarely use advanced temperature-displacement theories in the FEM formulation. Very often their consideration are restricted to the shell-nozzle connection or, vice versa, they focus on the main shell. Nowadays a complex FEM analysis with non-linear material behavior and advanced thermal approach is possible thanks to a proper software and increasing computational power of a hardware.

There are at least two general issues that should be checked on this matter:stress distribution in the nozzle-shell intersection,reduction of the shell thickness of the pressure vessel.

The authors of this paper performed such analyses using the ABAQUS [39] finite element method software.

## 2. Materials and Methods

The authors considered a pressure vessel designed for reaction gases under internal design pressure equal to 2.5 MPa and hydrotest pressure equal to 3.9 MPa. The maximal design temperature was set to 150 °C. The vessel consists of a cylindrical shell (1200 mm of internal diameter) and two parabolic dish heads. The total length of the vessel (excluding nozzles) is 2750 mm. It is equipped with inlet and outlet nozzles, lifting trunnions, and is welded to supporting saddles. The geometry of the vessel is presented in Figure 1. Please note that two variants of the vessel are shown, namely without any reinforcing pads and with elliptical pads.

The elements of the vessel are made of 1.4541 steel. Mechanical properties of steel were assumed according to codes [40,41,42,43,44,45]. Poisson’s ratio of steel is equal to 0.3, and Young’s modulus is 188.94 GPa. An elastic-plastic material model was assumed with the use of a classical von Mises yield criterion (PLASTICITY in the Abaqus code [39]) with yield stress equal to 200 MPa (with corresponding strain equal to 0.2%) and tensile strength of 500 MPa (with ultimate strain equal to 40%). Due to the elevated temperature of the stored gases, the numerical model of the vessel took into account a change of mechanical properties of steel in the maximum design temperature, i.e., at 150 °C. According to the codes [40,41,42,43,44,45], these properties are as follows: yield stress 167 MPa, tensile strength—410 MPa. To recreate thermal stress distribution, a coupled thermal-stress analysis in Abaqus software was assumed. This analysis demands a few specific material constants, which were given as follows: conductivity—15 Wm^−1^K^−1^, expansion coefficient—1.6× 10^−5^ K^−1^, specific heat—500 Jkg^−1^K^−1^.

Two different load combinations were pondered in the FEM analysis:Case 1—deal load, working pressure 2.5 MPa with temperature equal to 150 °C and loading of nozzles,Case 2—deal load, test pressure 3.9 MPa, and loading of nozzles.

Results presented in the paper were divided into these two cases.

The cylindrical shell thickness varied depending on whether reinforcing pads were used. In case of the lack of pads, the thickness was assumed as 28 mm, which resulted in a total mass of the vessel equal to 6.11 t. The use of the elliptical pads allowed for a thickness equal to 18 mm, and the total mass decreased to 4.26 t. The thickness of the pads was also assumed to be 18 mm.

The vessel was meshed using 10-node quadratic 3D tetrahedral elements, and the average mesh size was equal to 5 cm. Meshing of the vessel is shown in Figure 2. The average finite element size was established after a mesh-sensitivity analysis, which showed that further decreases in the finite element size did not cause any significant changes in results. Individual parts of the vessel were constrained to each other using the “Tie” constraint option in Abaqus software. In a real vessel, the reinforcing pads are welded to the shell. For the sake of simplicity of the 3D FEM model, welding lines were not implemented in the model. In the production process of this vessel, rigorous defectoscopic inspection of welds is planned, and the bearing capacity of the welds must be no less than the bearing capacity of the connected parts; hence, the assumption that any damage will occur earlier in the shell or in the nozzle than in the weld. Boundary conditions were applied at the bottom of each saddle, one of them defined as a pinned support, and the other as a roller support, where displacement is allowed along a symmetry axis of the cylindrical shell.

## 3. Results and Discussion

The presentation of results of FEM calculations is divided into two different load combinations: case 1 and case 2 (described in the previous chapter). The results are in the form of maps of von Mises stress and resultant displacements. A specific exponential notation used in the Abaqus software needs to be explained; for instance, e+08 means 10 to the power 8, while e-03 is equal to 10^−3^. First of all, the maximal stress should not exceed the tensile strength of steel. Concentration of stress is expected in nozzle-shell or nozzle-pad-shell connections, and that is why the authors of this paper decided to show stress distribution not only in the main shell of the vessel, but also in the connections.

### 3.1. Case 1—Working Pressure with Elevated Temperature

Von Mises stress distribution in the shell and in the selected connections (i.e., in these connections, where stress reached its maximal value) is presented in Figure 3, Figure 4, Figure 5 and Figure 6. In both cases (with and without reinforcing pads), the maximal stress appeared in the connection between the shell and the nozzles and/or the nozzles and the reinforcing pads. Apart from the connections, stress was lower than 160 MPa, which is a smaller value than the yield stress of steel at 150 °C. The maximal stress in the connections reached 300 MPa in the variant with the elliptical reinforcing pads, but it appeared in only one finite element (see Figure 5). Stress distributions in both cases are comparable, but keep in mind that in the case of the version with the reinforcing pads, the shell is 10 mm thicker.

The resultant displacements of the vessel are presented in Figure 7 and Figure 8. The maximal displacement was smaller than 10 mm in both variants and appeared at the end of the vessel equipped with the roller support. These values of the resultant displacements are relatively small (even if the temperature reached 150 °C) and therefore, yielding of the steel in the main shell is not expected. Once again, results are comparable, so the variant with the reinforcing pads provides good results with a lower thickness of the shell.

### 3.2. Case 2—Test Pressure

The results obtained for the second load combination are presented in the same order as in Section 3.1. In Figure 9, Figure 10, Figure 11 and Figure 12, we can see maps of von Mises stress, and the last two figures (Figure 13 and Figure 14) present the resultant displacement of the vessel. The maximal stress reached 268 MPa, but this time not only in the connections, but also in the dish heads (see Figure 9). In case of the higher thickness of the shell, the maximal stress was concentrated in the connections only (Figure 10).

The resultant displacements in both cases were very small–they did not exceed 23 mm. It is understandable because of the lack of thermal effects. The distributions of the resultant displacements were comparable in both variants, so the variant with the reinforcing pads is preferable.

The authors of the paper [34] decided to introduce the “stress concentration factor (SCF)”, which is a ratio of the maximal stress in the critical zone (for instance in the connection) to the nominal stress, produced over the cross section of this zone (for instance the shell without any discontinuities). In their work, they compared the SCF obtained in the cases of circular and elliptical reinforcing pads. The SCFs obtained for a vessel with the elliptical pads were higher than in the case of circular pads and generally increased with the growth of the nozzle’s diameter. The authors of this paper decided to apply a different approach, mainly because they simulated numerically the whole vessel (instead of a connection zone as presented in [34]). A simple ratio of the maximal stress to the tensile strength (at the given temperature) is enough to track a stress concentration and to give practical feedback for structural designers, as the defined ratio is simply equal to the efficiency ratio of the material. The analyzed vessel is equipped with five nozzles, but only three of them are connected to the main shell, and the maximal stress reached significant values there. The properties of the three nozzle-shell connections are listed in Table 1.

As we can see, in the variant with the reinforcing pads, the general tendency for the growth of the ratio with the growing nozzle diameter occurred. This observation is consistent with the results presented in [34]. In the variant without the pads, the ratio did not change significantly. However, one should bear in mind that in this variant, the main shell of the vessel is 10 mm thicker. The highest efficiency factor occurred in case 1 of the variant with reinforcing pads for the largest inner diameter of the nozzle, but still, the stress is lower than the ultimate stress. We can also see an important influence on the efficiency factor when the reinforcing pads are used. The factor has a stable value (in this study, it is around 0.50) despite the diameter of the nozzle. When no reinforcing pad is used, the factor can change its value significantly, taking the highest value for the largest diameter of the nozzle. These results show us that the efficiency ratio can also be a good tool to estimate the “sensitivity” of a nozzle-shell connection. More sensitive connections demonstrate larger changes in the factor with increasing diameter of the nozzle.

## 4. Conclusions

The analysis of the results presented in this paper allows us to draw the following conclusions:The use of the elliptical reinforcing pads leads to a significant decrease in the thickness of the shell. In the presented example, the decrease in thickness is equal to 10 mm (28 mm to 18 mm, 36%), and the total mass savings are equal to 1.85 t (6.11 t to 4.26 t, 30%).The abovementioned savings also lead to a decrease in the carbon footprint during the production of the vessel.The concentration of the maximal von Mises stress appears in the nozzle-shell connection, so the use of the reinforcing pads is recommended.The coupled thermal-displacement model in Abaqus software can be an effective tool to analyze pressure vessels at elevated temperatures.The introduced efficiency ratio grows with the growing nozzle diameter in the variant with the reinforcing pads, but the von Mises stress is lower than the ultimate stress. The ratio reaches its highest value in the largest nozzle under the first load combination and without reinforcing pads (0.735). The use of the reinforcing pads leads to an almost stable value of the efficiency ratio, regardless of the diameter of the nozzle. The factor can be useful in engineering practice to provide quick information on the effect caused by the reinforcing pads.

The presented work demonstrates the effectiveness of the nonlinear finite element model. The model couples static loading with thermal effects in the elastic-plastic range. Moreover, the mechanical properties of the material are defined as temperature-dependent. This model is able to recreate the realistic behavior of the vessel. As presented in the Introduction, only Jamadar et al. [8] decided to perform a nonlinear finite element analysis of a vessel with reinforcing pads. The vast majority of scientists (as shown in the Introduction) and structural engineers, however, choose the linear-elastic model. This model may be true if one is sure that the von Mises stress is lower than the yield stress of the material, especially in the case of increased temperature and in the nozzle-shell connections.

The presented work shows good efficiency of the elliptical reinforcing pads, which allow a large decrease in the thickness of the main shell. The elliptical shape of the reinforcing pads may raise some doubts, especially in terms of the complexity and cost of their production compared to “classic” circular pads. In fact, the elliptical pad requires less material (in comparison with the circular one) as the material distribution in the pad is oriented in one particular dimension–in this case, along the height of the cylindrical shell of the vessel. Less material means less waste when using a metal sheet. Moreover, the shape of the circular pad is no less complicated than the elliptical one, as it should fit perfectly into the cylindrical shell of the vessel. Nowadays, CNC machines allow precise production of such elements, and mass, standardized, thought-out, and optimized production of the elliptical pads will allow for additional savings.

A future work of the authors contains a detailed analysis of a carbon footprint and energy savings thanks to the use of the elliptical pads. Moreover, the authors are going to compare elliptical pads with circular ones and analyze more advanced issues, for instance, the performance of the vessel over time (fatigue and crack growth). The methodological side of the work will be enriched with the use of adaptive meshing, which is available in the Abaqus software.

## Figures and Tables

**Figure 1 materials-18-02318-f001:**
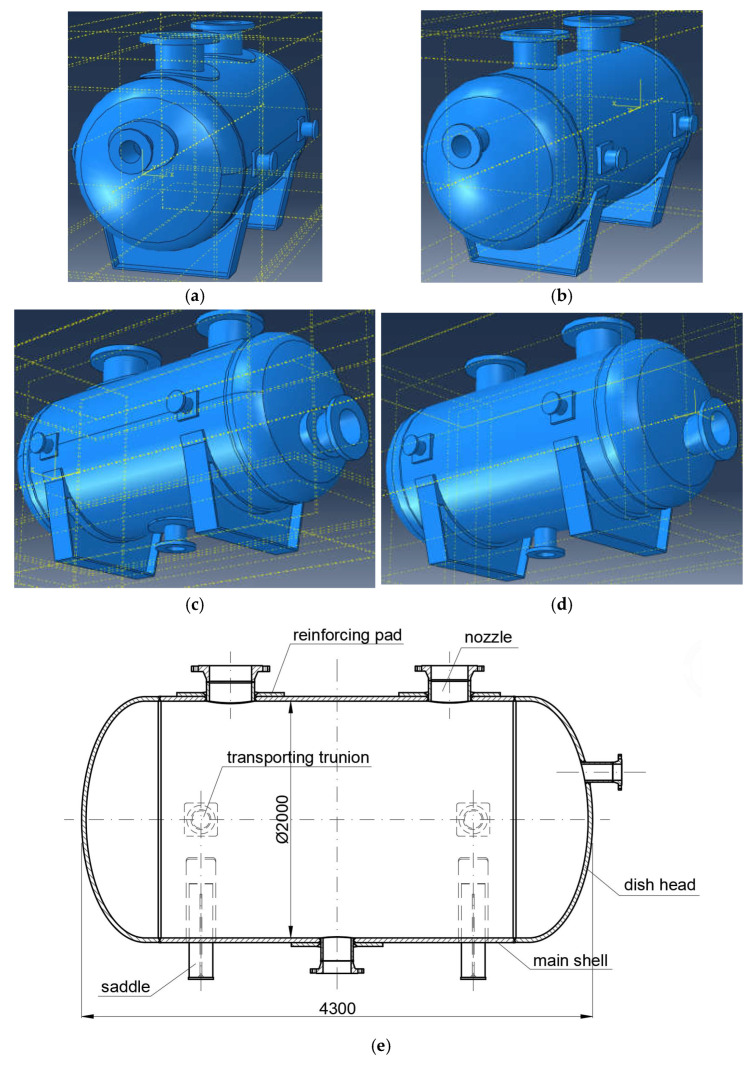
3D views of the vessel (**a**,**c**)—with elliptical reinforcing pads, (**b**,**d**)—without pads; (**e**) 2D section of the vessel with reinforcing pads.

**Figure 2 materials-18-02318-f002:**
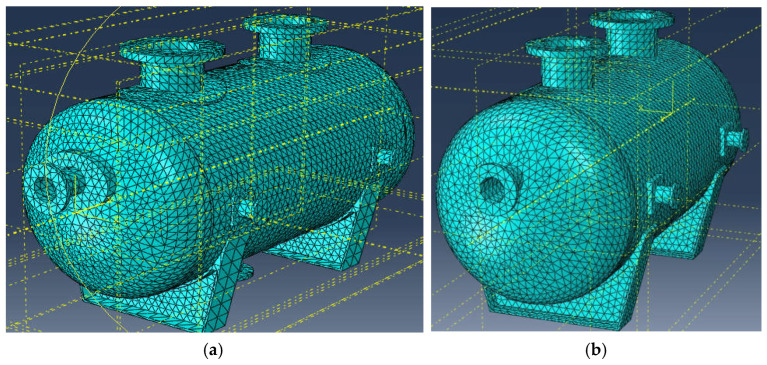
Meshing of the vessel (**a**) with elliptical reinforcing pads, (**b**) without pads.

**Figure 3 materials-18-02318-f003:**
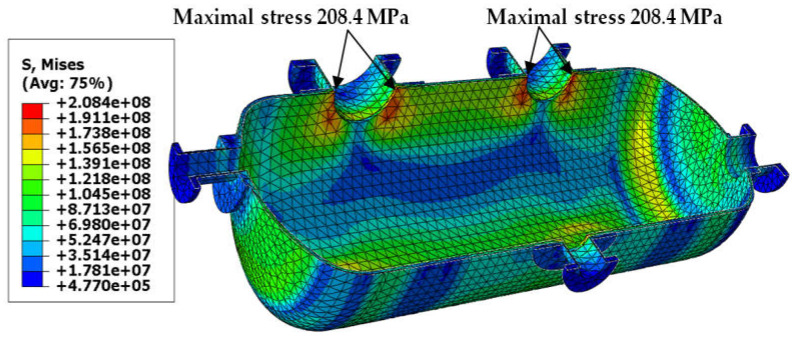
Von Mises stress in [Pa]–vessel with elliptical reinforcing pads, load combination—case 1.

**Figure 4 materials-18-02318-f004:**
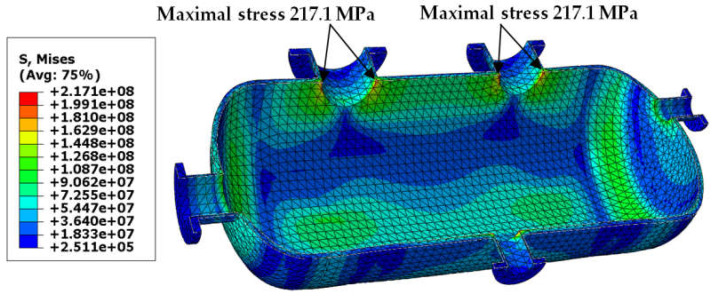
Von Mises stress in [Pa]–vessel without reinforcing pads, load combination—case 1.

**Figure 5 materials-18-02318-f005:**
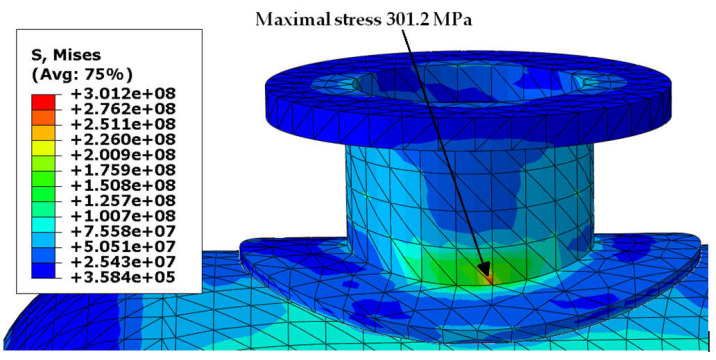
Von Mises stress in [Pa]–nozzle-pad connection, load combination—case 1.

**Figure 6 materials-18-02318-f006:**
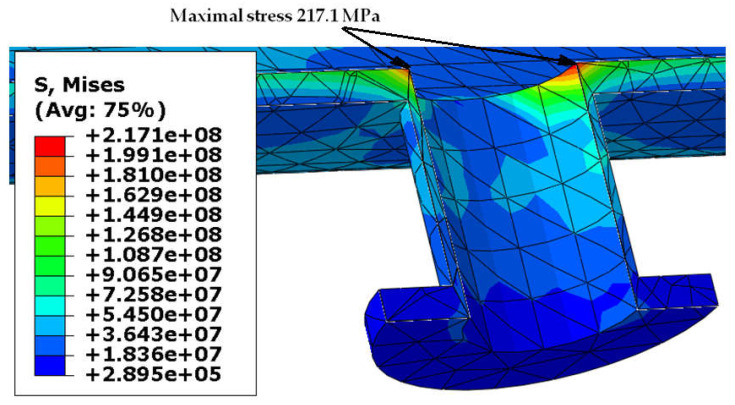
Von Mises stress in [Pa]–nozzle-shell connection (no pads), load combination—case 1.

**Figure 7 materials-18-02318-f007:**
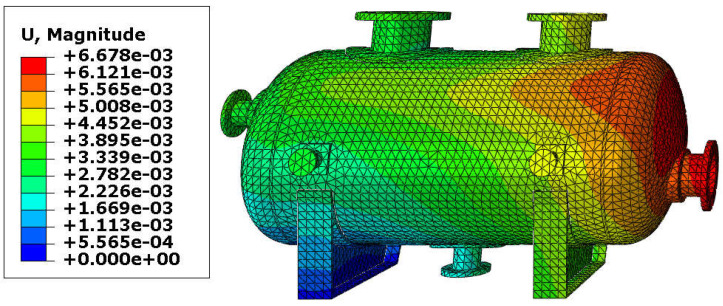
Displacement in [m]–vessel with elliptical reinforcing pads, load combination—case 1.

**Figure 8 materials-18-02318-f008:**
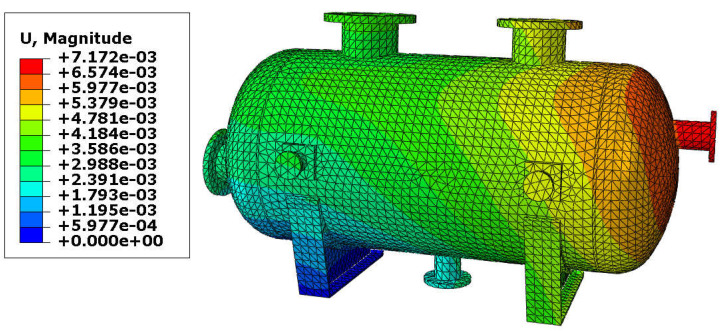
Displacement in [m]–vessel without reinforcing pads, load combination—case 1.

**Figure 9 materials-18-02318-f009:**
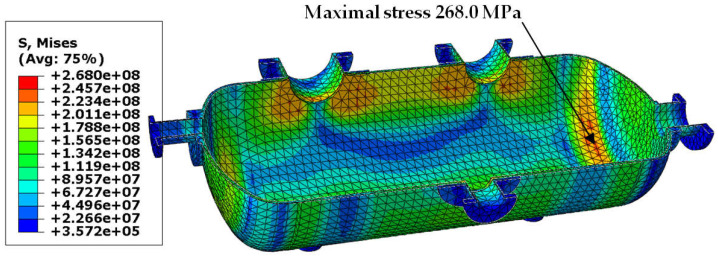
Von Mises stress in [Pa]–vessel with elliptical reinforcing pads, load combination—case 2.

**Figure 10 materials-18-02318-f010:**
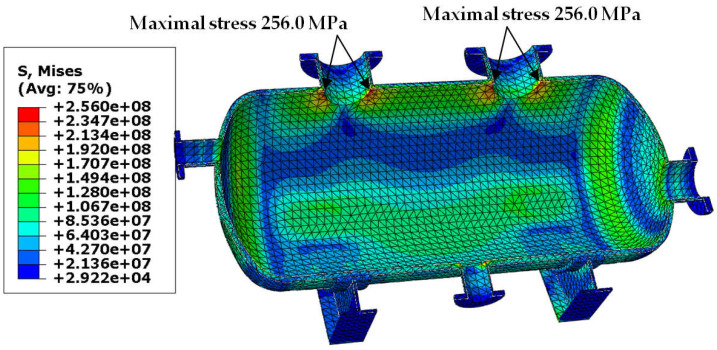
Von Mises stress in [Pa]–vessel without reinforcing pads, load combination—case 2.

**Figure 11 materials-18-02318-f011:**
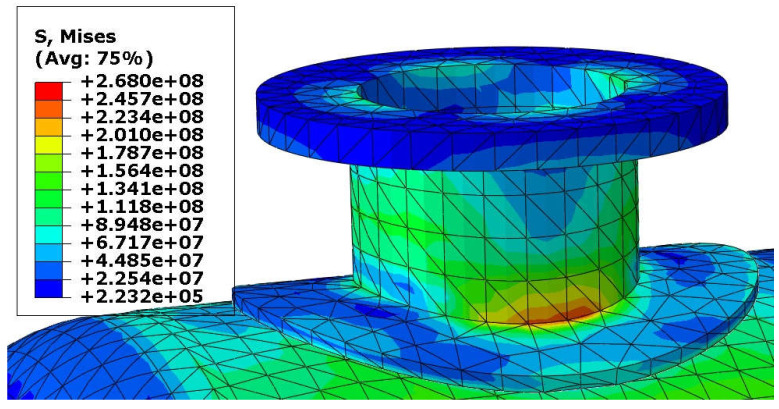
Von Mises stress in [Pa]–nozzle-pad connection, load combination—case 2.

**Figure 12 materials-18-02318-f012:**
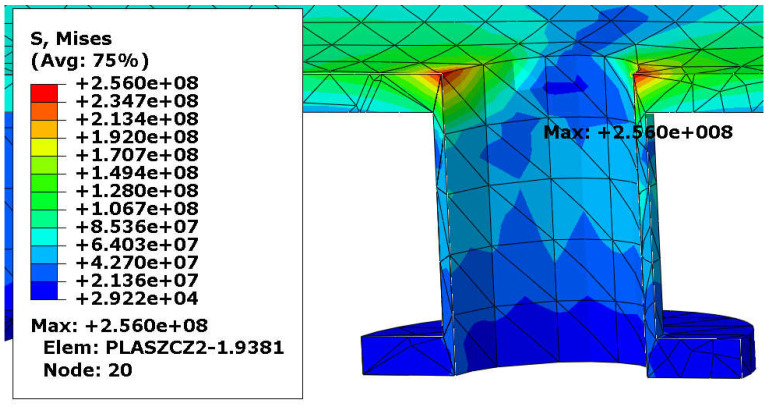
Von Mises stress in [Pa]–nozzle-shell connection (no pads), load combination—case 2.

**Figure 13 materials-18-02318-f013:**
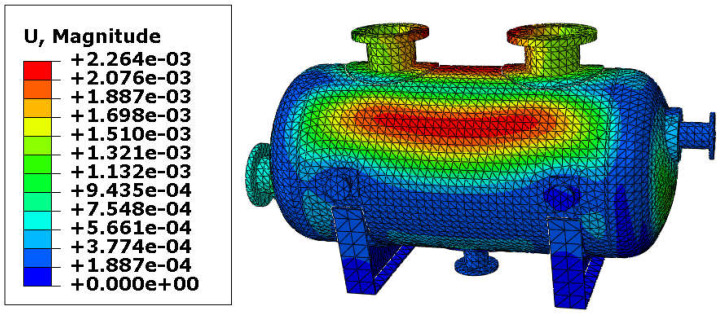
Displacement in [m]–vessel with elliptical reinforcing pads, load combination—case 2.

**Figure 14 materials-18-02318-f014:**
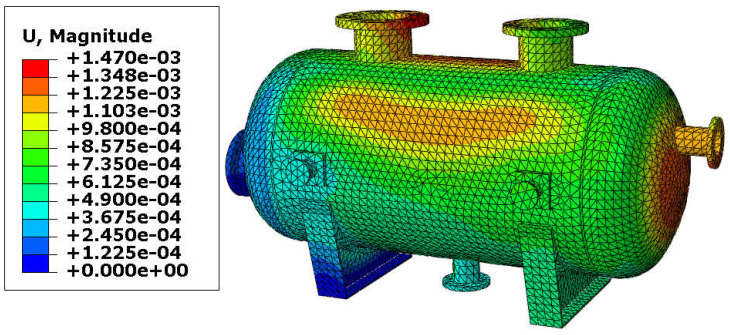
Displacement in [m]–vessel without reinforcing pads, load combination—case 2.

**Table 1 materials-18-02318-t001:** Efficiency ratios of shell-nozzle connections.

Inner Diameter of Nozzle [mm]	Variant	Load Combination	Maximal Stress [MPa]	Tensile Strength [MPa]	Efficiency Ratio
333.6	with reinforcing pad	Case 1	301.2	410	0.735
303.9	with reinforcing pad	Case 1	270.6	410	0.660
152.3	with reinforcing pad	Case 1	133.5	410	0.326
333.6	without pad	Case 1	199.9	410	0.488
303.9	without pad	Case 1	200.0	410	0.488
152.3	without pad	Case 1	217.1	410	0.530
333.6	with reinforcing pad	Case 2	268.0	500	0.536
303.9	with reinforcing pad	Case 2	266.2	500	0.532
152.3	with reinforcing pad	Case 2	179.4	500	0.359
333.6	without pad	Case 2	245.7	500	0.491
303.9	without pad	Case 2	243.1	500	0.486
152.3	without pad	Case 2	256.0	500	0.512

## Data Availability

No new data were created.

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
