# Peer review of "Numerical Study of the Reinforcing Pads Geometry of Pressure Vessels"

_materials, 2025, doi:10.3390/ma18102318_

Round 1
Reviewer 1 Report
Comments and Suggestions for Authors
Comments and Suggestions for Authors (will be shown to authors)
1. Line 26: Check the phrase. Try to remove 'interesting'. Line 27: what are 'states'?
2. It is a good practice done by the author to list previous (similar) work in the introduction. It would be better to add a critical review for each of those (e.g., advantages/disadvantages) and how those works are linked together.
3. Figure 1: it would be better to label each part, such as 'pads', 'nozzle', and 'weld'. Try to include '2D' CAD graph as well.
4. Since there is stress concentration at the 'connections' (pad/nozzle or nozzle/shell), please clarify if any denser mesh has been applied in those regions. If yes, please describe it in detail.
5. Please clarify the materials of 'enforcing pads'. Are they welded to the shell? If yes, where are the welding points/lines? Any consideration of welding microstructure since they have very different mechanical responses to the bulk materials?
6. Considering a rather simple geometry of the component, please try to rationalize your conclusions using simple analytical forms in the 'Discussion' section.
7. It would be a better practice to list the materials models in the paper, even though those have been documented in the codes (ref. 37-42).
Author Response
The authors would like to thank to the Reviewer 1 for their valuable comments and suggestions. Below there are authors' responses.
Comment 1: Line 26: Check the phrase. Try to remove 'interesting'. Line 27: what are 'states'?
Response 1: The phase has been checked and corrected, "interesting" has been removed. The word "states" means ultimate and serviceability limit states (ULS and SLS).
Comment 2: It is a good practice done by the author to list previous (similar) work in the introduction. It would be better to add a critical review for each of those (e.g., advantages/disadvantages) and how those works are linked together.
Response 2: a critical summary of the cited works has been complemented in the "Introduction" section.
Comment 3: Figure 1: it would be better to label each part, such as 'pads', 'nozzle', and 'weld'. Try to include '2D' CAD graph as well.
Response 3: the parts of the vessel have been labelled and the CAD drawing has been added (as Figure 1e).
Comment 4: Since there is stress concentration at the 'connections' (pad/nozzle or nozzle/shell), please clarify if any denser mesh has been applied in those regions. If yes, please describe it in detail.
Response 4: yes, an optimal mesh (average size of a finite element equal to 5 cm) was established after an analysis of mesh-sensitivity in Abaqus software. For the mesh average size equal to 2,5 cm (a two times finer mesh) the maximal stress chenged from 268.0 MPa (5 cm mesh) to 257.8 MPa, which is a bit lower than 4% of relative change. After this analysis, the presented vessel was meshed with 5 cm mesh to make all the analysis (case 1 and case 2 of loading combinations) less time-consuming (please keep in mind, that the whole model was meshed with 3D elements with one additional degree of freedom devoted to temperature, so the total size of the FEM model is very large even for 5 cm element size).
A proper comment on the finite element size has been added in the text.
Comment 5: Please clarify the materials of 'enforcing pads'. Are they welded to the shell? If yes, where are the welding points/lines? Any consideration of welding microstructure since they have very different mechanical responses to the bulk materials?
Response 5: the reinforcing pads were modelled with the same material as the whole vessel. The pads are welded to the shell. For the sake of simplicity of the 3D FEM model, welding lines were not implemented in the model. In the production process of this vessel, rigorous defectoscopic inspection of welds is planned and the bearing capacity of the welds must be no less than the bearing capacity of the connected parts, hence the assumption that any damage will occur earlier in the shell or in the nozzle than in the weld.
A proper comment on the welding has been added in the text.
Comment 6: Considering a rather simple geometry of the component, please try to rationalize your conclusions using simple analytical forms in the 'Discussion' section.
Response 6: a proper modification of the "Discussion" section has been made.
Comment 7: It would be a better practice to list the materials models in the paper, even though those have been documented in the codes (ref. 37-42).
Response 7: a detailed description of material properties was given in the "Materials and methods" section, in the text just below the Figure 1. Material models (elasto-plastic material model, coupled thermal-stress analysis) were also invoked in this section.
Reviewer 2 Report
Comments and Suggestions for Authors
The abstract summarizes the work’s scope and contribution well. To further improve it, the authors should add specific numerical outcomes (e.g., percentage thickness reduction or mass savings) to clarify the practical impact of using elliptical pads.
The approach is technically robust. Nevertheless, the absence of a parametric sensitivity study on pad dimensions and limited geometric diversity slightly limits the applicability of the conclusions.
-
Parametric Study of Elliptical Pad Dimensions
→ Reason: To identify optimal geometric ratios (major/minor axis, thickness) that minimize stress concentrations or maximize weight savings. -
Comparison with Circular Pads on the Same Model
→ Reason: While the introduction discusses circular pads, no direct numerical comparison is conducted on the same vessel geometry. -
Fatigue or Lifecycle Assessment
→ Reason: The study focuses only on static stress. Including fatigue life or crack growth potential would reflect real-world vessel performance over time. - Figures and tables are generally well-structured and clearly presented. The contour plots (von Mises stress, displacement) effectively demonstrate differences between the two configurations. Figure captions are concise but would benefit from more descriptive annotations (e.g., labeling nozzle zones with peak stress).
Visuals are useful and of high quality. The clarity of interpretation would improve if critical zones were marked or annotated directly on the figures.
The conclusion is solid and aligned with the presented results. A brief comment on future work would enhance the practical outlook.
The manuscript effectively highlights a novel gap—the full-vessel simulation with elliptical pads—and justifies the relevance of this specific research.
References are relevant and up-to-date. Consider adding more recent citations (e.g., 2023–2024) on reinforcing pad optimization or nonlinear vessel analysis to further strengthen the literature basis.
-
Original: "Thickness of the cylindrical shell differed depending on the variant of the reinforcing pads."
Revised: "The cylindrical shell thickness varied depending on whether reinforcing pads were used." -
Original: "The numerical model of the vessel was meshed with 10-node 3D quadratic tetrahedrons..."
Revised: "The vessel was meshed using 10-node quadratic 3D tetrahedral elements..." -
Original: "Because of elevated temperature of stored gases..."
Revised: "Due to the elevated temperature of the stored gases..." -
Original: "No yield of steel in the main shell of the vessel is expected."
Revised: "Yielding of the steel in the main shell is not expected." -
Original: "Resultant displacement of the vessel is presented in the Figures..."
Revised: "The resultant displacements of the vessel are shown in Figures..."
Author Response
The authors would like to thank to the Reviewer 2 for their valuable comments and suggestions. Below there are authors' responses.
Comment 1: The abstract summarizes the work’s scope and contribution well. To further improve it, the authors should add specific numerical outcomes (e.g., percentage thickness reduction or mass savings) to clarify the practical impact of using elliptical pads.
Response 1: a proper line has been added to the abstract.
Comment 2:
The approach is technically robust. Nevertheless, the absence of a parametric sensitivity study on pad dimensions and limited geometric diversity slightly limits the applicability of the conclusions.
-
Parametric Study of Elliptical Pad Dimensions
→ Reason: To identify optimal geometric ratios (major/minor axis, thickness) that minimize stress concentrations or maximize weight savings. -
Comparison with Circular Pads on the Same Model
→ Reason: While the introduction discusses circular pads, no direct numerical comparison is conducted on the same vessel geometry. -
Fatigue or Lifecycle Assessment
→ Reason: The study focuses only on static stress. Including fatigue life or crack growth potential would reflect real-world vessel performance over time.
Response 2: all the above mentioned suggestions are valuable and will be considered by the authors in their future work. The rewieved paper is a case study on a specific vessel, i.e. with elliptical pads (vs the case without the pads). As mentioned in the "Acknowledgement" section, the presented work is a part of a scientific report, prepared as part of research and development (B+R) work of Kielce University of Technology. The commisioned study concerned only elliptical pads under static load with no optimization requirements. In other words, the geometry of the pads were given as an input.
Comment 3: Figures and tables are generally well-structured and clearly presented. The contour plots (von Mises stress, displacement) effectively demonstrate differences between the two configurations. Figure captions are concise but would benefit from more descriptive annotations (e.g., labeling nozzle zones with peak stress).
Visuals are useful and of high quality. The clarity of interpretation would improve if critical zones were marked or annotated directly on the figures.
Response 3: labels have been added in the figures. Moreover, the authors supplemented the paper with a 2D "CAD" drawing (Figure 1e)
Comment 4: The conclusion is solid and aligned with the presented results. A brief comment on future work would enhance the practical outlook
Response 4: future work of the authors has been supplemented in the "Conclusions" section
Comment 5: References are relevant and up-to-date. Consider adding more recent citations (e.g., 2023–2024) on reinforcing pad optimization or nonlinear vessel analysis to further strengthen the literature basis.
Response 5: three new references on pressure vessels have been added in the text.
Comment 6:
-
Original: "Thickness of the cylindrical shell differed depending on the variant of the reinforcing pads."
Revised: "The cylindrical shell thickness varied depending on whether reinforcing pads were used." -
Original: "The numerical model of the vessel was meshed with 10-node 3D quadratic tetrahedrons..."
Revised: "The vessel was meshed using 10-node quadratic 3D tetrahedral elements..." -
Original: "Because of elevated temperature of stored gases..."
Revised: "Due to the elevated temperature of the stored gases..." -
Original: "No yield of steel in the main shell of the vessel is expected."
Revised: "Yielding of the steel in the main shell is not expected." -
Original: "Resultant displacement of the vessel is presented in the Figures..."
Revised: "The resultant displacements of the vessel are shown in Figures..."
Response 6: all the suggested revisions have been applied in the text.
Round 2
Reviewer 1 Report
Comments and Suggestions for Authors
The author improved the manuscript well based on my previous comments. I have a minor comment on the numerical side. I suggest using an adaptive meshing scheme where a smaller mesh is used at the joints, etc. This can significantly improve accuracy without much increase in computational cost.
Author Response
Comment 1: "I have a minor comment on the numerical side. I suggest using an adaptive meshing scheme where a smaller mesh is used at the joints, etc. This can significantly improve accuracy without much increase in computational cost."
Response 1: The authors would like to thank the Reviewer 1 for their constructive suggestion. We are going to apply an adaptive meshing in our future work, espacially since this option is also available in Abaqus software. An additional comment has been put in the text in the "Conclusions" section (future work of the authors).